# Accuracy Assessment of the GlucoMen^®^ Day CGM System in Individuals with Type 1 Diabetes: A Pilot Study

**DOI:** 10.3390/bios12020106

**Published:** 2022-02-09

**Authors:** Daniel A. Hochfellner, Amra Simic, Marlene T. Taucher, Lea S. Sailer, Julia Kopanz, Tina Pöttler, Julia K. Mader

**Affiliations:** Division of Endocrinology and Diabetology, Department of Internal Medicine, Medical University of Graz, 8036 Graz, Austria; daniel.hochfellner@medunigraz.at (D.A.H.); amra.ajsic@medunigraz.at (A.S.); marlenetaucher@gmail.com (M.T.T.); lea.sailer02@gmail.com (L.S.S.); juliakopanz@gmx.at (J.K.); tina.poettler@medunigraz.at (T.P.)

**Keywords:** diabetes technology, CGM, accuracy, type 1 diabetes, sustainability

## Abstract

The aim of this study was to evaluate the accuracy and usability of a novel continuous glucose monitoring (CGM) system designed for needle-free insertion and reduced environmental impact. We assessed the sensor performance of two GlucoMen^®^ Day CGM systems worn simultaneously by eight participants with type 1 diabetes. Self-monitoring of blood glucose (SMBG) was performed regularly over 14 days at home. Participants underwent two standardized, 5-h meal challenges at the research center with frequent plasma glucose (PG) measurements using a laboratory reference (YSI) instrument. When comparing CGM to PG, the overall mean absolute relative difference (MARD) was 9.7 [2.6–14.6]%. The overall MARD for CGM vs. SMBG was 13.1 [3.5–18.6]%. The consensus error grid (CEG) analysis showed 98% of both CGM/PG and CGM/SMBG pairs in the clinically acceptable zones A and B. The analysis confirmed that GlucoMen^®^ Day CGM meets the clinical requirements for state-of-the-art CGM. In addition, the needle-free insertion technology is well tolerated by users and reduces medical waste compared to conventional CGM systems.

## 1. Introduction

The introduction of continuous glucose monitoring (CGM) represents one of the most important advancements within diabetes treatment and self-management over the last decades. CGM provides easy access to current glucose levels, glucose trends, and the retrospective analysis of glucose excursions, thus facilitating easier and better diabetes management for both people living with diabetes (PLWD) and health care professionals. As a result, PLWD using CGM technology show improvement in HbA1c, glucose variability, hypoglycemia prevalence, overall well-being, and treatment satisfaction and have less fear of hypoglycemia compared to SMBG [1,2,3].

One critical issue with this technology remains the quality of the accuracy of glucose measurements, even though vast improvements in this regard have become evident over the last years [4,5,6,7]. Nevertheless, most current CGM have already reached accuracy levels of SMBG and are, therefore, labeled for nonadjunctive use by regulators, meaning that CGM can be utilized for treatment decisions without the subsequent SMBG confirmation [8,9,10,11]. Furthermore, as CGM enables easy assessment of the time spent in target range (TIR), which has been shown to be a valid marker of glycemic control alongside HbA1c, there has been an evolution of individual treatment guidelines for CGM use with a focus on TIR [12]. However, further improvement in CGM accuracy, particularly in the hypoglycemic range, is crucial for the development of reliable diabetes technology, especially for automated insulin delivery systems, as well as for reducing the burden of diabetes management for PLWD.

Another key aspect of CGM technology that is crucial to adherence is comfort in both wearing and inserting the sensor. Registry data show that wearing discomfort is the prevailing factor for CGM discontinuation [13]. Therefore, new developments in diabetes technology aim to reduce discomfort in diabetes management and, thus, facilitate better quality of life with diabetes. 

As CGM technology is made available to a growing number of PLWD, the negative impact on the environment and natural resources increases due to the use of disposable products and accumulation of plastic and medical waste, including hazardous parts such as insertion needles. This problem is increasingly being addressed not only by environmental organizations but also by users of diabetes technology and patient organizations. Therefore, the Diabetes Technology Society started its Green Diabetes Initiative to promote the development of medical devices in terms of sustainability and reduce the environmental impact associated with advancements in diabetes technology [14]. 

In the present analysis, we aimed to assess the accuracy and usability of a novel CGM system that is the first to feature predominately reusable components with needle-free insertion.

## 2. Materials and Methods

In this monocentric, open-label, non-randomized, single-arm clinical study, eight individuals with type 1 diabetes were equipped with two sensors of the GlucoMen^®^ Day CGM system (Waveform Cascade, A. Menarini Diagnostics, Florence, Italy; Figure 1) to wear at home. This novel CGM features a needle-free insertion system containing a single-use, disposable sensor, the reusable transmitter to be recharged after 14 days of use (for up to 5 years), the reusable sensor insertion tool (for up to 5 years), the transmitter charging unit with a USB port, and the GlucoMen^®^ Day App supported by Android and iOS devices allowing for viewing glucose levels and trends, for calibrating the system, and for setting alarms. 

The sensors were simultaneously inserted into the subcutaneous adipose tissue on the opposite sides of the lower abdomen and worn for 14 days. Additionally, participants were requested to calibrate the CGM devices once daily and perform 8–9 finger-prick glucose tests per day (GlucoMen Day METER, A. Menarini Diagnostics, Florence, Italy). 

The primary objective was to assess the device’s accuracy compared to a laboratory reference instrument (YSI 2300, Yellow Springs, OH, USA). The secondary objectives included comparing CGM to SMBG and a precision assessment by evaluating the agreement between the two sensors worn in parallel. Additionally, a usability assessment including a 10-item questionnaire with an ordinal rating scale asking about insertion pain, adhesive adherence, problems with calibration, reliability of readings, usability of and satisfaction with the mobile application, device wearability, and overall experience was performed.

### 2.1. Meal/Insulin Challenge

On days 4 and 10 of the study, a 5-h meal and insulin challenge was performed at the research center. Each participant consumed a standardized meal containing 100 g of carbohydrates and received an increased insulin bolus (the regular bolus insulin dose plus 20%) subcutaneously. Venous plasma and capillary blood samples were collected simultaneously every 20 min. Plasma glucose was measured on site with YSI, while SMBG was performed using the study-specific BG meter.

### 2.2. Data Analysis

CGM accuracy was assessed by calculating MARD, MAD (mean absolute difference), Median ARD, Median AD, and CEG for both CGM/YSI and CGM/SMBG matched pairs. Following a frequently used approach in glucose monitoring system accuracy evaluations, MARD (and median ARD), as a percentage error, was calculated for glucose above the 100 mg/dL threshold only (i.e., values in the 100- to 400-mg/dL range), while, for the lower glucose values (ranging from 40–99 mg/dL), it calculated the absolute error as MAD (and median AD) [15,16]. The lag time between CGM and blood glucose data was determined for each sensor and applied prior to calculating MARD and MAD. The lag time was calculated as the shift in time that provided the best CGM/reference correlation, using the Poincarè method (time shift giving the best R 2 vs. references), and was adjusted for SMBG and YSI data and for YSI data individually for each sensor. The average lag time was calculated as the weighted average (based on the respective N of data) of the lag time of all the sensors. All statistical analyses were performed following the intention-to-treat principle.

## 3. Results

Eight adult participants (3 females (37.5%), age 41.6 ± 13.3 years, BMI 28.0 ± 6.1 kg/m^2^, HbA1c 55.6 ± 12.2 mmol/mol, diabetes duration 13.9 ± 6.5 years) completed the study. During the study period, on average, 94.4% of the theoretically possible data was collected and used for analysis after applying the exclusion criteria for data recording (estimated CGM signal).

### 3.1. Accuracy CGM vs. YSI

Overall, the 450 CGM/YSI matched pairs available for analysis were generated from glucose data collected within the range of 40–400 mg/dL. This resulted in a MARD of 9.7 (±9.4)% and a MAD of 20.5 (±18.7) mg/dL, as summarized in Table 1.

By analyzing specific glucose ranges, it was observed that sensor accuracy as assessed by MARD was better in the 201–400-mg/dL range compared to the 100–200-mg/dL range (6.1% vs. 10.7%), while MAD was lower in the 40–70-mg/dL range compared to the 71–99-mg/dL range (19.5 vs. 20.9 mg/dL). Furthermore, both MARD and MAD were lower on day 4 compared to day 10 (7.4% vs. 11.4% and 17.7 vs. 27.3 mg/dL). CEG analysis showed that 84.9% of CGM/YSI data pairs was in the clinically acceptable zone A, while the combined percentage for zone A and B was 97.8. CEG is displayed in Figure 2a.

### 3.2. Accuracy CGM vs. SMBG

During sensor use at the research center and at home, a total of 1957 CGM/SMBG matched data pairs were collected in the range of 40–400 mg/dL. The overall MARD was 13.1 (±12.8)% for values between 100–400 mg/dL, while the MAD for glucose values between 40–99 mg/dL was 16.6 (±16.8) mg/dL, as summarized in Table 1. Similar to what was observed for YSI, the MARD and MAD were lower in the higher glucose range (13.4% for 100–200 mg/dL vs. 12.2% for 201–400 mg/dL and 20.2 mg/dL for 40–70 mg/dL vs. 15.2 mg/dL for 71–99 mg/dL), while the CEG analysis showed a combined percentage of 98.2 for zones A and B (Figure 2b).

### 3.3. Sensor Precision

Glucose values recorded by the two sensors worn in parallel were evaluated using mean ± SD and coefficient of variation (CV) for each participant. An average SD of 11.1 mg/dL and an average CV of 9.7% were observed, demonstrating acceptable agreement between the two sensors.

### 3.4. Usability

The participants were requested to complete a questionnaire containing 10 questions with an ordinal five-scale answer rating. When asked about pain perception at insertion, 50% stated that the procedure was painless and 25% claimed it to be less painful than finger pricking. The remaining 25% found the insertion as painful as routinely used SMBG (12.5%) or more painful than finger pricking (12.5%). There was also predominant satisfaction with sensor adhesive, wearability, calibration procedure, and user-friendliness of the dedicated mobile application. The results of the usability assessment can be found in the Appendix A.

## 4. Discussion

High CGM accuracy is crucial for good diabetes management, especially when CGM readings are used for insulin dosing both alone or in combination with open/closed-loop systems. Accurate readings are also essential for precise calculations of time in range (TIR), a parameter that has become increasingly popular for treatment decisions and the overall assessment of glycemic control. By modeling the 7-point glucose profiles collected in the DCCT (Diabetes Control and Complications Trial), it was shown that TIR was strongly associated with development of microalbuminuria and retinopathy progression, suggesting that it represents a valid surrogate indicator of microvascular complications [17].

So far, MARD has been commonly used to assess overall CGM accuracy, while MAD is used to assess accuracy in the low glycemic range. Good CGM performance is generally assumed in systems with an overall MARD < 10%, although analytical performance cannot be fully assessed by a single parameter [18]. In the present analysis, the GlucoMen^®^ Day CGM achieved a MARD of 9.7% compared to YSI during meal/insulin challenges (with one daily calibration), while compared to SMBG, a MARD of 13.1% was obtained over the 14-day wear period. The higher MARD calculated for SMBG may result from lower accuracy associated with glucose meters compared to laboratory reference. This finding is often seen during assessment of CGM devices at home [19,20,21]. The results of the present analysis are comparable to those obtained for other CGM systems such as the real-time (rt) CGM G6 (Dexcom, San Diego, CA, USA) with a MARD of 9.9% or the intermittently scanned (is) CGM systems Freestyle Libre 1 and 2 (Abbott, Chicago, IL, USA) with respective MARDs of 12.0% and 9.2% in adults compared to YSI [22,23,24].

Furthermore, similar to earlier findings in CGM systems, there was an increase in MARD/MAD in the low glycemic range while accuracy was higher in hyperglycemic ranges [25,26,27]. Since this effect was more pronounced during the meal challenges in our analysis, it can be assumed that the assessed CGM system is more accurate following postprandial glucose excursions. Interestingly, MAD was slightly lower in the 40–70-mg/dL range compared to the 71–99-mg/dL range, suggesting an increased or at least stable accuracy in relevant hypoglycemia, whereas other CGM systems show an impaired accuracy when BG levels drop below 70 mg/dL [22,25,27]. Given the fact that most CGM systems are prescribed due to impaired hypoglycemia awareness, CGM accuracy in hypoglycemia is essential for giving a timely warning and taking adequate countermeasures to avoid hypoglycemia.

In addition to MARD/MAD as measures of numerical accuracy, clinical accuracy, as expressed by CEG, should also be evaluated to support the sensor performance data and should be favored over Clarke Error Grid or Continuous Error Grid analysis [15]. In our study, the CEG analysis showed 98% of all data pairs in the clinically relevant zones A and B for both reference methods, thereby verifying the clinical accuracy of GlucoMen^®^ Day. For comparison, the CEG analysis of Freestyle Libre 1 and 2 showed a respective 99.7% and 99.9% of matched pairs in zones A and B [23,24,25].

Advancements in diabetes technology have so far drastically decreased the burden for people with diabetes. The reduction of painful procedures associated with diabetes management improved quality of life and facilitated good glycemic control [28,29,30]. The replacement of SMBG and introduction of CGM in pediatric patients with diabetes was a crucial advance in reducing finger pricking. However, especially in young children, sensor insertions are reported to be one of several barriers of this technology regardless of the significant advances of CGM use for children and parents in diabetes management [31]. Further reduction of painful procedures may lead to increased adherence in CGM users, which is known to be age-dependent and especially poor in adolescents [32]. In the present analysis, the CGM system with needle-free insertion intended for ages 6 and above resulted in high user satisfaction in the majority of users claiming the insertion procedure to be painless or less painful than finger pricking. An important innovative feature of GlucoMen^®^ Day CGM is reusability of most components, resulting in reduced environmental impact. The wish for reducing the ecological footprint of diabetes technology is frequently expressed by users and patient organizations, especially in the face of the present climate crisis. Common CGM insertion systems often include bulky, disposable applicators that contribute to a substantial increase in plastic and potentially hazardous medical waste accompanied by a significant amount of packaging waste. This represents a rising problem worldwide and is fueled by the, per se, favorable, increasing availability of CGM. Looking at data from the DPV (Diabetes Patienten Verlaufsdokumentation) and the T1D exchange registry, CGM use in type 1 diabetes increased from 6% in 2011 to 38% in 2018 [33]. In absolute numbers, CGM use is presumably even higher due to its prevalence in type 2 diabetes and is expected to grow due to increasing reimbursement by healthcare providers. The growing ecological burden associated with diabetes technology is already acknowledged by the scientific community and is subject to ongoing discussion [14,34]. GlucoMen^®^ Day CGM includes a rechargeable transmitter and a reusable needle-free sensor insertion applicator, both of which can be used for up to 5 years. Due to reduced insertion pain and reusable components, this and similar systems can help promote sustainable diabetes technology and stimulate other positive developments in this sector, such as biodegradable components. Therefore, regulatory bodies should aim to establish rules for design and development of diabetes technology with reusable components that meet medical needs and are environmentally friendly. Additionally, waste reduction should be accounted for in cost-effectiveness calculations and reimbursement strategies for such products.

## 5. Conclusions

The present analysis suggests that the GlucoMen^®^ Day CGM is a user- and environmentally friendly system that meets the current clinical requirements for state-of-the-art CGMs. The reduced ecological impact of this needle-free system has to be emphasized and may support further advances in sustainable diabetes technology.

## Figures and Tables

**Figure 1 biosensors-12-00106-f001:**
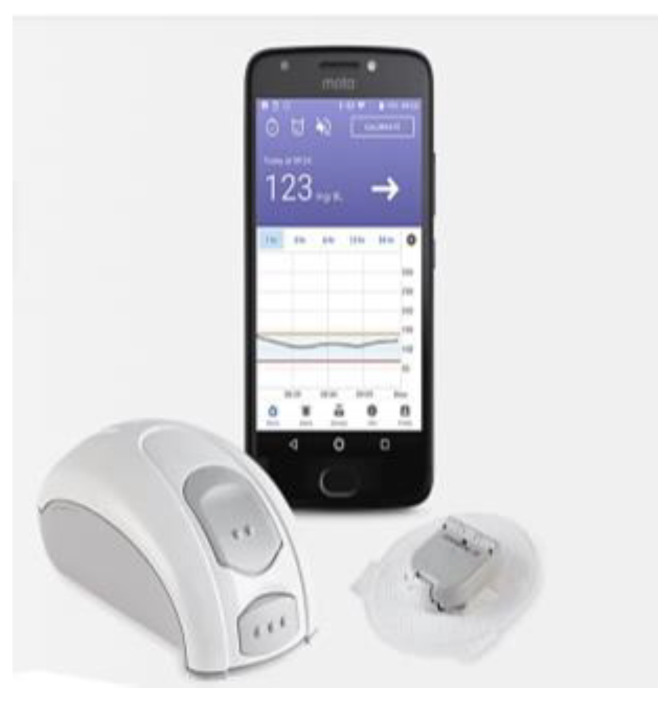
The GlucoMen^®^ Day CGM system.

**Figure 2 biosensors-12-00106-f002:**
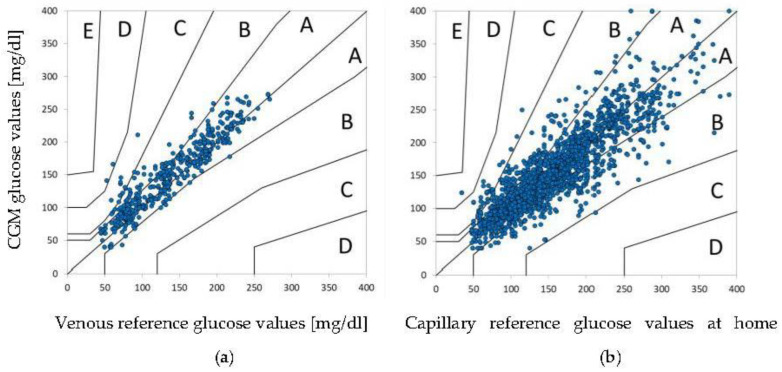
(**a**,**b**) CEG during study phases. (**a**) CGM compared to venous reference glucose values during meal challenge at the research center; (**b**) CGM compared to capillary glucose values at home.

**Table 1 biosensors-12-00106-t001:** System accuracy compared to YSI and SMBG.

YSI		SMBG	
**[Glucose] < 100 mg/dL**		**[Glucose] < 100 mg/dL**	
**MAD** (±SD), mg/dL	20.5 (+/−18.7)	**MAD** (±SD), mg/dL	16.6 (+/−16.8)
**Median AD** [IQR 25th/75th], mg/dL	16.5 [9.5–24.0]	**Median AD** [IQR 25th/75th], mg/dL	12.0 [3.0–23.0]
**[Glucose] ≥ 100 mg/dL**		**[Glucose] ≥ 100 mg/dL**	
**MARD** (±SD), mg/dL	9.7 (+/−9.4)	**MARD** (±SD), mg/dL	13.1 (+/−12.8)
**Median ARD** [IQR 25th/75th], mg/dL	6.7 [2.6–14.6]	**Median ARD** [IQR 25th/75th], mg/dL	9.8 [3.5–18.6]

## Data Availability

The data presented in this study are available on reasonable request from the corresponding author. The data are not publicly available due to company policy of data privacy so that competitors cannot directly assess the full data set.

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
