# Peer review of "Accuracy Assessment of the GlucoMen® Day CGM System in Individuals with Type 1 Diabetes: A Pilot Study"

_biosensors, 2022, doi:10.3390/bios12020106_

Round 1

Reviewer 1 Report

Thank you very much for giving me the opportunity to review this work. Mader et al present data from a pilot study on a new continuous glucose monitoring device with needle-free insertion. These are important new findings and add impact to the literature as all prescribable CGM devices at the moment are either inserted with a needle or placed subcutaneously in a small operation. The paper is well written and the results are clearly presented.

I suggest the following edits to further increase the quality of the manuscript:

It has not been disclosed if all of the participants in this trial were adults. Please add this missing information and discuss potential age-related benefits and challenges of this specific device (e.g. implications for elderly people and children). 

Reviewer 2 Report

The current manuscript contains important information about a novel CGM system.  I think the manuscript is well written- it is also important to present-publish information about novel CGM devices.

However, I think that is very important to have the following information for this new system, in order to have this manuscript suitable for publication:

-It is true that median values should be used in the lower/hypoglcyemic range. However Mean ARD (MARD) is also reported, acknowledging again that Median is most appropriate for this range. I would like the authors to add this information (MARD for 40-99 mg/dl)

-In addition to MAD it is important to add the Median ARD

- I would like the authors to have this information in tables.  This will make very clear to the reader. The figures are helpful however the tables will contain the most important information.

- Can the authors provide some information about the fact that the device is painful (12.5% of responders) or as painful with POC (12.5%)?

-Regarding the above qs a description of the CGM device is missing. Can they also add a picture of it, if possible?

Reviewer 3 Report

The idea is original, however, the data presented is few and not enough.
Furthermore, I suggest that the authors in the discussion section propose the advantages of the equipment based on current references.
the conclusion section is very short and the authors do not suggest a proposal

Round 2

Reviewer 2 Report

.

Reviewer 3 Report

I appreciate the authors for the corrections made in the manuscript entitled "Accuracy Assessment of the GlucoMen® Day CGM System in Individuals with Type 1 Diabetes: a Pilot Study".